# Remember rhythm and rhyme:

# Memorability of narratives for science communication

Aquiles Negrete

Centro de Investigaciones Interdisciplinarias en Ciencias y Humanidades

CEIICH-UNAM

aqny@unam.mx

*'Every man's memory is his private literature'*

Aldous Huxley

***Abstract***

Once upon a time narratives where considered to be a non-reliable way of representing and communicating science. Nowadays, narratives are widely accepted as an accurate way of conveying science, they represent an effective emotional trigger, a lasting memory structure and a powerful aid for learning. To study how memorable different ways of presenting information are is a fundamental task for science communication in order to evaluate materials that not only need to be understood by the general public, but also need to be retained in the long-term as a part of the communication process. In this paper I will give a brief introduction to cognitive psychology and the study of memory in relation to narratives.

Evidence from the field of memory studies suggests that narratives represent a good recall device. They can generate emotion and this in turn is a way of focusing attention, promoting rehearsal in memory and inducing long-term potentiation. Similarly, a story produces semantic links that might assist in storing and retrieving information from memory. Studies suggests that memory span and paired associate recall have implications in storing and recalling narratives. Evidence also suggests that the use of stories as modelling tools can organise information, provide schemas and allow extrapolation or prediction. Finally literature on memory suggests that narratives have a value as mnemonic devices.

*1.  Introduction*

The question of how knowledge can be presented to the public in order to convey as much information as possible with maximum fidelity is a central one for science communication, (Dornan, 1990; Durant *et al.,* 1989). Memory is one possible way of assessing learning (Sternberg, 2003), and therefore of judging the successful communication of information. Studying how memorable different text formats are, represents a fundamental task for science communication in order to produce materials that are not only expected to be understood by individuals but also stored in the long-term memory.

Much of the information that we store in our memory is not acquired first hand through personal experience, but second hand, through reading or listening to other people talk about their experiences (Cohen, 1989). Memory for spoken information and memory for written information differ in important ways. Reading is a private and solitary occupation; it has no conversational context such as intention, intonation, gesture, facial expression, or personality of the speaker. Written material has to be more formally structured and must conform to certain rules and formats to be intelligible to a wide range of potential readers.

In general, we remember meaning better than wording (Cohen, 1989). The general rule for narratives (short stories, drama, comics, novels, etc.) appears to be that the meaning, the gist, the most important and most relevant facts are preserved by the memory (Cohen, 1989). Almost any material becomes easier to remember if it is included in a narrative (Bruner, 1986; Bruner 1990; Crowley, 2018; ElShafie, 2018). There are several factors concerning memory that make narrative a lasting structure, some of them related to the memory process itself and others to the intrinsic characteristics of narratives as a means of expressing information.

*2.  Objective and methodology*

The objective of this work is to provide a literary review of memory studies regarding memorability of narratives.

In previous work (Negrete, 2009; Negrete and Lartigue, 2010; Negrete 2013; Rios and Negrete 2013; Negrete, 2014; Lartigue and Negrete 2016) we provided empirical evidence suggesting that science can be communicated and learned through narratives and that this represents a more enjoyable way of learning compared to traditional texts. In particular, we found that narrative information is retained for lengthier periods than factual information in long-term memory. Our evidence suggested that narratives constitute an important means for science communication to convey information in an accurate, memorable and enjoyable way.

In this opportunity my aim is to examine what has been reported in literature regarding features of the memory process that contributes to make narratives a memorable device. Although narratives have implications in short memory processes, I will concentrate on long-term memory, the most relevant features for science communication.

## *3. Narrative representation*

A dominant model of rationality implies a single type of discourse, one that puts forward hypotheses, reported evidence and systematically inferred conclusions. Stories, in contrast, frequently carry the connotation of falsehood or misrepresentation (Bruner 1986). However, several authors acknowledge that many scientific and mathematical hypotheses emerge as little stories or metaphors. In Howard's view (1991) there is a relationship between science and storytelling. This author considers, for instance, that science represents an example of constructing meaning through storytelling.

Bruner originally proposed two modes of cognitive functioning: paradigmatic and narrative. Each provides a different way of organising experience, constructing reality and communicating knowledge. They are, at the same time, complementary and irreducible to one another. While paradigmatic knowledge is focused on what is common among items, narrative knowledge focuses on the particular and special characteristics of actions. Human action is the result of the interrelation of previous learning, experience, present and future expectation. While paradigmatic knowledge is carried in individual words that name a concept, narrative knowledge is maintained in stories with plot. Storied memories retain the

complexity of the situation in which an action was undertaken, and the emotional and
motivational meaning connected with it. The collection of storied experiences provides a
basis for understanding new action episodes by means of analogy (Amos and Wisniewski
96  1995).


Narratives can take different forms. Among the different types of narratives, parables and
myths have a particular interest for science communication. Both are aids in understanding
difficult concepts. Although the latter may not match our current sense of reality, they can be
used in science communication to analyse the values and limits of scientific knowledge
(Blades 2001). Also science fiction is of paramount importance in science communication as
it is the literary genre most frequently used to represent, explore and play with science.
Science can be used as the subject of the narrative, as the basis for the plot, as a background
or setting, or even as a metaphor (Willis 1998). Science fiction represents a valuable tool for
science education (Gough, 1993 and Appelbaum, 1995) and communication.
For this work a narrative is a particular type of discourse production, in which events and
actions are assembled in an organised unity with the help of an intrigue (Connelly and
Clandinin, 1990). Narrative texts answer the question "What happened?" Characters, events
and plot exist in a world where time goes by (Amos and Wisniewski, 1995). According to
the cognitive model, narratives can be seen as memory enhancing devices (Atkinson and
Shiffrin, 1971).

*4.  Literary review on memory studies*

4.1 Cognitive Psychology
Cognition is a sub-discipline of psychology that studies how humans perceive, learn,
remember and think about information (Sternberg, 2003). Memory is the means by which
humans retain and draw upon past experience and use this information in the present (Tulving
and Craik, 2000). It is the record of experience that underlies learning. Learning can be
defined as a biological mechanism that permits us to face a changing world, i.e., it is a process
by which long lasting changes in the behaviour potential take place as a result of experience.
In cognitive psychology three main memory operations are distinguished: (i) encoding, (ii)
storage, and (iii) retrieval (Baddeley, 2000). Each operation represents a stage in memory
processing. Through encoding the individual transforms sensory data into a form of mental
representation; through storage, the encoded information is maintained in the memory and
through retrieval, it is pulled out for use. Pioneering work by Tulving and Pearlstone (1966),
as well as Murdock (1961), suggested that although encoding, storage, and retrieval
phenomena are theoretically clearly defined, in practice there is considerable overlap and
they are therefore too interdependent to allow for working with each as a separate unit.

4.2 Long-term Memory

There are different ways of encoding in long-term memory (LTM) (Stenberg, 2003; Crowley,
2018). Most information stored in long-term memory seems to be semantically encoded.
There is evidence in early work on the area that other forms of encoding exist in long-term
memory, such as visual encoding (Frost, 1972) and acoustic encoding (Nelson and Rothbart,
1972), but they play a minor role in comparison to semantic encoding.

Information from short-term memory is transferred to long-term memory depending on
whether the information involves declarative (declarative knowledge refers to recalling facts)
or non-declarative memory. Some forms of non-declarative memory like priming and
habituation are ephemeral and dissipate rapidly; others such as procedural and conditioning
are maintained for longer periods, especially when rehearsed. For declarative knowledge to
enter into LTM, two main processes are involved: attention and association (of new
information with previous knowledge and also of schemas). The process of integrating new
information into stored information is referred as consolidation (Squire, 1986).

Retention and enhancement of memory during consolidation can be promoted with different
meta-memory strategies (Koriat and Goldsmith, 1996; Metcalfe, 2000). These strategies
involve a conscious act of reflection by rehearsing and organising (mnemonics) new
information destined to stay in long-term memory.

4.2.1 Long-term Potentiation and Rehearsal

Every experience leaves a trace in the brain. Every experience is potentially a memory but
only some traces seem to become permanently imprinted into brain tissue. Every experience
– whether it is a real or perceived event, a thought, a feeling, a fragment of the imagination,
or a recollection of a previous experience – involves the activation of a unique neural firing
pattern (Maren, 1999). Some events produce strong and long-lasting patterns, which tend to
recur continually. When connections are repeatedly activated, they form even more robust
links, which bind them into a single unit: long-term potentiation (LTP). Research suggests
that memories generated in this way (LTP) can last a lifetime (Barhrick & Hall, 1991).

Rehearsal is perhaps the simplest and most effective strategy that can be used in a memory
task. It is an interactive process by which information in short-term memory is continually
articulated or 'refreshed'. Its importance is that it maintains information in short-term
memory by ensuring a sufficiently high level of activation and it facilitates the transfer of
information to long-term memory and subsequent retrieval by allowing additional time for
more elaborate item processing (Dempster, 1981).

There are three important moments for Long-Term Potentiation (long lasting memory):
attention, emotional response and rehearsal. It is interesting noting that a typical oral joke
(normally the narrative of something funny happening to somebody) concentrate these tree
elements.  When someone is going to tell a joke people pay "attention" to the speaker. If the
joke is good, they "laugh" (emotional response). Hours later or even the next day, when
people remember the joke, they will laugh probably again (rehearsal). That is the way people
learn the jokes and reproduce them with friends and colleagues. The joke has a precise
structure in order to be funny. It is interesting how we are able to remember such structure
with remarkable fidelity so we are able to retell the joke with the precision required to make
people laugh. Humorous narratives should be considered as an important resource for science
communication as they represent a tool that can induce Long-Term Potentiation by
promoting attention, emotional response, and rehearsal (See for example Primo Levi´s
narrative in section 4.6).
4.2.2 Oblivion

Oblivion is defined as the decline of performance after learning. It occurs after a certain
period. To measure it, researchers observe behaviour after a period in which the learned
behaviour has not taken place (retention period).

It is worth noting that oblivion occurs quickly when we learn lists of unrelated words or
unsystematic items. In contrast, if the text is meaningful, it is more likely that we will
remember it for longer periods. Previous knowledge (proactive knowledge) can also reduce
oblivion (Squire, 1986). Pioneering work by Sir Frederick Bartlett (1932) showed that a story
which was difficult to understand was made modern and comprehensible by participants
thanks to proactive knowledge. His experiments consisted of presenting an indigenous, North
American story called *The War of Ghosts* to a group of participants in Britain. Bartlett found
that his participants distorted their recall to provide a story that was more comprehensive to
them. Their previous knowledge and expectations had a substantial effect on their
recollection. In so doing, Bartlett developed the idea that in memory tasks we use our already
existing schemas, which affect the way we recall and learn. In the geosciences context, it has
been suggested that Myths (a form of narratives) help in reducing oblivion of geological
hazards (flooding, eruptions and earthquakes) and this proactive knowledge has helped to
create a culture of prevention in different human groups (Barthes, 2013; Crowley, 2018;
Lanza and Negrete, 2007). One interesting example are the myths concerning exploding
lakes, as Lake Nyos in Cameroon. As Shanklin (2007) reports, many stories were based on
the assumption that lakes are the homes of ancestors and spirits and can be source of death.

4.3 Emotion and Attention

Experiencing emotion provides a basis for simple learning and memory (Sternberg, 2003).
Emotional learning and memory such as fear conditioning are simple forms of associative
learning that supports the acquisition of knowledge; it is acquired rapidly and retained over
long periods (Maren, 1999). An effect of emotional stimulation is to direct attention towards
the events that provoked it. This attention in turn augments the brain activation associated
with the event. Attention is effectively the first stage of laying down memory (Rupp, 1998).
Evidence shows that what distinguishes enduring experiences from those that are lost is that
when they occurred they either created or coincided with higher than normal levels of
emotion (Baddeley, 1997). It is clearly vital for humans to remember events that are
emotionally arousing because they are likely to be important ones. They can be used to guide
present and future actions. They can be used, for example, to avoid danger (as geological
hazards) or to steer us towards a desirable outcome (O'Brien, 2000). Interestingly, the same
neuro-chemicals that are released into the bloodstream to put the body on alert also instruct
the brain to store a lasting record of the moment. This is the case for acetylcholine,
noradrenaline, dopamine and glutamate, which all participate in the creation of links between
neurons (Rupp, 1998; Zak 2007).

Durability of a particular memory seems to depend on how exciting the original experience
was (or how excited the individual's brain was when it occurred), how much attention was
paid to it and how often it is recalled (Stenberg, 2003). In Lotman's words (1990), 'narratives
are a way of expressing ideas and amplifying emotions'. If emotions are generated, there is
greater opportunity to concentrate attention and produce long-term potentiation are higher.
Also, the possibility to rehearse the emotions is greater, since we tend to repeatedly remember
passages that result from a meaningful or emotional experience (Stenberg, 2003).
4.4 Memory in Context and Knowledge Networks

According to Gough (1993), context is of paramount importance in order to understand
memory process. No subject exists in isolation. Knowledge does not remain neatly
compartmentalised into disciplines, but spills over and 'transgresses' boundaries. Everything
that happens has a context, not only circumstances and surroundings but also internal states,
emotions and physical feelings. If an event is laid down as a memory, some of its context is
laid down with it and becomes a hook for remembering (Rupp, 1998). Contextual elements
can be valuable aids to recall because when one part of a memory is retrieved, it often 'hooks
out' all the others.
Memories that have similar connotations, forming links based on meaning, are called
semantic links. Semantic links act like a cross-referencing system: once we have found a
useful piece of information, we can connect it with many more that might also be relevant
(Cohen, 1989). Memories that are formed simultaneously are linked by association. These
associative links are fundamental to our understanding of the world and often allow us to
make predictions based on previous experience (see also section on *Models and Schemas*).
Most of the time, semantic and associative links work unconsciously: as soon as one concept
is activated in memory, activation spreads automatically to other ideas related through
meaning or past experiences.
A story can be seen as an expressive device that by means of a plot associates characters,
situations, places, and information to produce semantic links and a cross-referencing system
that can assist in storing and retrieving information in, and from, memory (i.e. scientific
knowledge).
4.5 Human Memory Systems
According to Tulving (1972), there are six major human memory systems: semantic,
episodic, procedural, perceptual, representational and short-term memory. There is
reasonable evidence of the existence of the first two types: semantic and episodic memory.
With the aid of semantic memory, individuals are able to register and store information about
the world in the broadest sense (i.e. not personally experienced) and are capable of retrieving
it. Semantic memory allows people to think about things that are absent to the senses at the
time (Tulving 1972). Semantic memory is automatic, i.e., it does not require a conscious
recollection. It develops earlier in childhood than episodic memory (Tulving 1972).
4.6 Episodic Memory
This is the type of memory used to remember events in our lives. Therefore, episodic memory
is related to the self-experiences in subjective space and time. An episodic memory consists
of memories that come from different areas of the brain that are bound together to create an
'episode', rather than a collection of impressions or items of knowledge (Crowley, 2018). In
contrast with semantic memory and other kinds of memory systems, in this case the
individual is able to transport into the personal past and future at will (Tulving 2000). In
times of crisis the individual is able to bring the past to the forefront in order to reinterpret
the events of a lifetime.

Tulving (1966) pointed out that retrieving information from each memory system is
associated with distinct memory awareness experiences. According to this author, when an
individual uses episodic memory, they are conscious of remembering past experiences,
whereas in the case of semantic memory, a person's conceptual knowledge is characterised
by memory awareness involving feelings of familiarity or "just knowing".

Episodic memory is characterised by two aspects of temporal structuring: the location of the
event in a specific past time in relation to the present and a temporal sequencing within the
episode remembered (Nelson 1972). Both of these aspects rely on a sense of the "extended
self" and apparently the role of autobiographic memory is to provide a sense of continuity of
the self across time from past to future (Nelson 1972).

There is a strong link between episodic memory and emotions. The way in which memories
are formatted determines their emotional significance and the retrieval pathways to other
episodic memories. Earlier experiences tend to be recalled from a third person's point of view
(i.e. as an observer), while more recent events are usually recalled from the first person's
point of view (i.e. as a participant). Emotions are usually stronger when memories are
recalled from a participant's point of view, while the observer's point of view tends to be
more objective.

Narratives offer information that is contextualised in real-life situations (episodes). When an
episode in a narrative work evokes emotion in the reader, this incident may become
memorable. Narratives (fictional or non-fictional) provide the opportunity to create episodes.
If the narrative episode evoke emotions and part of it contain science, then it would be
reasonable to expect that information contained in it (included science) will form a lasting
memory.
The following narrative is a shortened version (performed by the author) of Primo Levi´s
"Nitrogen" (1985). It provides an example of an episode that includes science and has proved
to be a memorable device (Negrete 2009).

*The client explained to me that he was the owner of a cosmetics factory and he wanted to*
*produce a certain kind of lipstick. He needed a few kilos of alloxan. He would pay a good*
*price for it, provided I committed myself by contract to supply it only to him. He had read*
*that alloxan in contact with the mucous membrane confers on it an extremely permanent red*
*colour, because it is not a superimposition, in short a layer of varnish like lipstick, but a true*
*and proper dye, as used on wool and cotton. I gulped, and to stay on the safe side replied*
*that we would have to see: alloxan is not a common compound nor very well known, I don't*
*think my old chemistry textbook devoted more than five lines to it, and at that moment I*
*remembered only vaguely that it was a derivative of urea and had some connection with uric*
*acid. I dashed to the library at the first opportunity and hastened to refresh my memory as to*
*the composition and structure of alloxan.*

*Alloxan is a hexagonal ring of oxygen, carbon, hydrogen and nitrogen; it is a pretty*
*structure! It makes you think of something solid, stable, well linked. In fact it happens also*
*in chemistry as in architecture that "beautiful" edifices, that is, symmetrical and simple, are*
*also the most sturdy: in short, the same thing happens with molecules as with the cupolas of*
*cathedrals or the arches of bridges. Alloxan was known for almost seventy years, but as a*
*laboratory curiosity: the preparation method described had a pure academic value, and was*
*made from expensive raw materials which (in those years right after the war) it was*
*optimistic to hope to find on the market. The sole accessible preparation was the oldest: it*
*did not seem too difficult to execute, and consisted in an oxidising demolition of uric acid.*
*Just that: uric acid, the stuff connected with gout, intemperant eaters, and stones in the*
*bladder. It was a decidedly unusual raw material, but perhaps not as prohibitively expensive*
*as the others.*

*Subsequent research taught me that uric acid, very scarce in the excreta of man and*
*mammals, constitutes, however, 5O percent of the excrement of birds and 90 percent of the*

*excrement of reptiles. Fine. I phoned the client and told him that it could be done, he just had*

*to give me a few days' time: before the month was out I would bring him the first sample of*

*alloxan, and give him an idea of the cost and how much of it I could produce each month.*

*The fact that alloxan, destined to embellish ladies' lips, would come from the excrement of*

*chickens or pythons was a thought which didn't trouble me for a moment. The trade of chemist*

*teaches you that matter is matter, neither noble nor vile, infinitely transformable, and its*

*proximate origin is of no importance whatsoever. Nitrogen is nitrogen, it passes miraculously*

*from the air into plants, from these into animals, and from animals to us; when its function*

*in our body is exhausted, we eliminate it, but it still remains nitrogen, aseptic, innocent. We*

*-I mean to say we mammals- who in general do not have problems about obtaining water,*

*have learned to wedge it into the urea molecule, which is soluble in water, and as urea we*

*free ourselves of it; other animals, for whom water is precious (or it was for their distant*

*progenitors), have made the ingenious invention of packaging their nitrogen in the form of*

*uric acid, which is insoluble in water, and of eliminating it as a solid, with no necessity of*

*having recourse to water as a vehicle.*

*I returned home that evening and informed my wife that the next day I would leave on a*

*business trip: that is, I would get on my bike and make a tour of the farms on the outskirts of*

*town in search of chicken shit. She did not hesitate, she would come along with me. But she*

*warned me not to have too many illusions: finding chicken shit in its pure state would not be*

*so easy. In fact it proved quite difficult. First of all, the pollina—that's what the country*

*people call it, which we didn't know, nor did we know that, because of its nitrogen content,*

*it is highly valued as a fertiliser for truck gardens—the chicken shit is not given away free,*

*indeed it is sold at a high price. Secondly, whoever buys it has to go and gather it, crawling*

*on all fours into the chicken coops and gleaning all around the threshing floor. And thirdly,*

*what you actually collect can be used directly as a fertiliser, but lends itself badly to other*

*uses: it is a mixture of dung, earth, stones, chicken feed, feathers, and chicken lice, which*

*nest under the chickens' wings. In any event, paying not a little, labouring and dirtying*

*ourselves a lot, my undaunted wife and I returned that evening with a kilo of sweated-over*

*chicken shit.*

*The next day I examined the material: there was a lot of gangue, yet something perhaps could*
*be got from it. But simultaneously I had an idea; just at that time, in the Turin subway gallery*
*an exhibition of snakes had opened: Why not go and see it? Snakes are a clean species, they*
*have neither feathers nor lice, and they don't scrabble in the dirt; and besides, a python is*
*quite a bit larger than a chicken. Perhaps their excrement, at 90 percent uric acid, could be*
*obtained in abundance, in sizes not too minute and in conditions of reasonable purity. This*
*time I went alone: my wife is a daughter of Eve and doesn't like snakes. The director and the*
*various workers attached to the exhibition received me with stupefied scorn. Where were my*
*credentials? Where did I come from? Who did I think I was showing up just like that, as if it*
*were the most natural thing, asking for python shit? Out of the question, not even a gram;*
*pythons are frugal, they eat twice a month and vice versa; especially when they don't get*
*much exercise. Their very scanty shit is worth its weight in gold; besides, they—and all*
*exhibitors and owners of snakes—have permanent and exclusive contracts with big*
*pharmaceutical companies. So get out and stop wasting our time. I devoted a day to a coarse*
*sifting of the chicken shit, and another two trying to oxidise the acid contained in it into*
*alloxan. The virtue and patience of ancient chemists must have been superhuman, or perhaps*
*my inexperience with organic preparations was boundless. All I got were foul vapours,*
*boredom, humiliation, and a black and murky liquid which irremediably plugged up the*
*filters and displayed no tendency to crystallise, as the text declared it should. Best to return*
*among the colourless but safe schemes of inorganic chemistry.*

4.7 Mnemonics

Before the invention of writing, and long afterwards in many cultures, stories were sung or
recited from memory. Rhythm, rhyme and melody were used to provide a framework that
aided in their memorisation. Mnemonics are one tool employed to aid recitation from
memory. It is defined as the art of improving memory, or a system to aid the memory, i.e.,
any strategy that helps people remember. It normally means signals for learning that will later
induce the experience to be remembered (Stenberg, 2003).

According to Lotman (1990), mnemonics can be seen as a way of internal communication
that is made up of messages to the self with the purpose of retaining information and includes
different sorts of memoranda and reminders. Essentially, such reminder devices add meaning
(or personal meaning) to otherwise meaningless, unrelated or arbitrary lists of items for the
individual. Mnemonics superimposes an artificial, logical structure (which can be seen as a
model) on data, which are not necessarily related. A mnemonic device can be an image
(*Alloxan is a hexagonal ring of oxygen, carbon, hydrogen and nitrogen; it is a pretty*
*structure! It makes you think of something solid, stable, well linked. In fact it happens also*
*in chemistry as in architecture that "beautiful" edifices, that is, symmetrical and simple, are*
*also the most sturdy: in short, the same thing happens with molecules as with the cupolas of*
*cathedrals or the arches of bridges*), an acronym, a verse, a rhyme (*matter is matter, neither*
*noble nor vile*), a peg word, a catch phrase or a story that helps us to remember (Luria, 1986).
In Yates' view (1992), a feature of Cosmas Rossellius´s book (*Thesaurus artificiosae*
*memoriae*) are the mnemonic verses given to help memorize orders of places in Hell, or the
order of the signs of the zodiac. These verses were written by Dominican inquisitor. These
carmina by the Inquisitor constitute an interesting example of the use of artificial memory
via mnemonics (Yates, 1992).
Most of the world's great religions have strong oral traditions in which sacred texts are
memorised in their entirety for prayer and to preserve them for posterity. For example, in the
*Mishna*, the Jewish written record of the oral law, some literary resources such as metaphors,
digressions and poetic images can actually be viewed as mnemonic aids. The *Qur'an* also
contains mnemonic aids. This religious book was written both as a work of rhythmic prose
and as an epic poem; thus, rhythm, rhyme, and meaning connect every word making it
memorable (Luria, 1986).
Narratives can be seen as mnemonic structures that superimpose an artificial, logical structure
on data which is not necessarily related. In this way scientific factual information can be
communicated by being embedded in a mnemonic structure (the story) which facilitates
future recollection.
4.8 Memory Span and Paired Recall Association

In early work in this area, Dempster (1981) defined memory span as the maximum length of
a series of words, images or items that can be reproduced at different stages in time. One of
the most practical and important implications of memory study is in education. As short-term
memory span is indicative of overall intellectual ability it can be used as a diagnostic tool
both for helping educators (and communicators) to adapt teaching (and learning materials)
to the specific needs of the learner and for measuring improvements in intellectual ability
Dempster (1981). Higher spans are the result of grouping and organisation (Estes, 1974).
Organisation, in turn, is one of the key elements of paired recall association.

Pioneering work by Epstein, Rock and Zuckerman (1960), suggested that when two objects
have been perceived or imagined to be interacting, recalling the name of one, in response to
the name of the other, is more frequent than when the objects have been perceived or
imagined to be side by side. This effect in memory is called paired recall association. As a
result of the relationship between two objects, they develop certain properties and
interactions. A relation or interaction constitutes a feature that characterises both objects,
which enables the individual to retrieve one when the other is provided (Wilton, 1989). An
interesting example, of the effectiveness of rhythm and paired recall association as mnemonic
aids is clear when we try to remember the lyrics of a song and it suffices to recollect its
rhythm in order to do so.

When words are used as units of meaning, the semantic components of the words are
activated (Wilton, 1990). If two words are associated semantically, this assures that common
structures are activated in that task. Therefore, in the search for recall, the items to be recalled
are found together. On the other hand, when words are used as a collection of symbols
without semantic meaning, the common structures are not activated and recall is
disorganised.

Following this line of argument, it would be plausible that stories represent a means of
increasing memory span, a way to facilitate retrieval from memory by paired recall
association and a powerful device to convey science to the general public in a long lasting
way.

4.9 Models

According to the classical work by Giere (1979), models can be classified into three
categories: scale models that represent reality to a particular scale; analogue models which
are useful for understanding other proposed new models; and theoretical models, the most
abstract form of a model as they are imaginary and often explained with analogical models.
Examples of the latter are the though experiments. A thought experiment is an idealisation
or abstraction of existing physical conditions. A thought experiment implies the use of visual
imagery abstracted from phenomena that we have actually experienced. This imagery allows
intuition, an impression of how things are connected, innovation and the possibility of
modelling in the mind. This kind of thinking was used by famous scientists such as Galileo,
Einstein, Maxwell, Bohr and Heisenberg.

In Casti's (1993) view, models can serve three purposes: they can be predictive, explanatory
and prescriptive. Prescriptive models give us the opportunity not just to explain or predict
but also to manipulate some aspect of the world through 'handles' on the model ( *op.cit*.).

Casti (1993) compares modelling with painting and other artistic disciplines. When an artists
paint, they never creates on canvas the exact image of the subject in front of them. Instead
they try to capture the essence of meaningful characteristics so that the viewer is able to know
more about the object painted than from looking at the real thing. In this sense the object art
(paint, sculpture, music, or literature) shows hidden characteristics by using magnifying
glasses, special lights, tones, rhythms or narrative resources. Giere´s (1979) and Casti's
(1993) arguments claim that stories can be seen as narrative models that have the power to
explain, the capacity to show scale, an ability to predict the future, to produce analogies and
metaphors as well as to theorise.
Yuri Lotman (1977) suggested that semiotic systems are models that explain the world in
which we live. Amongst all semiotic systems, language is the primary modelling system in
which we apprehend the world by means of the model that it provides. Myth, cultural rules,
religion, paint, music, literature (narratives) and science are secondary modelling systems.
All of them are of equal interest as models to understand and talk about the world.

In Johnson-Laird's (1983) words: '. . . stories are represented as mental models in the reader's
mind". To construct a mental model of a story is to imagine what was happening in the
narrative. A mental model is a global representation that integrates information from different
parts of the story. It is constructed as the story unfolds, and represents the scene, characters,
and events, incorporating spatial, temporal, and casual relations (Johnson-Laird, 1983).
Mental models have the intuitively appealing feature of treating memory for stories and
memory for real-world events as essentially the same (Yates, 1992).

Narratives can also be seen as secondary modelling systems in which information is
represented and organised by means of a plot. This enables us to make sense of reality and
prepare information in an organised structure ready for future recall. Stories can be seen as
narrative models as they depict the model which has the capacity to explain. For example in
the capacity to show scale as in Carbon by Primo Levi, the possibility to show in few pages
processes that take millions of years as in *The Crabs Take Over the Island* by Anatoly
Dnieprov (1966), an ability to predict the future as in *The Time Machine* by H.G. Wells
(1895), or to produce analogies and metaphors as in *Flatland* by Edwin A. Abbot (1884) and
to theorise as in Italo Calvino's *Cosmicomics* (1969). Using narratives provide a powerful
tool to communicate Science.

4.10 Story schemas

One of the earliest studies of memory and narratives was carried out by Frederic Bartlett
(1932). Unlike many psychologists of his day, Bartlett recognised the need to study memory
retrieval with connected texts rather than studying unconnected strings of digits, words or
nonsense syllables. He introduced the idea that schemas, or mental frameworks, built up from
prior knowledge and experience, are influential in determining and shaping the memory of a
story (see section 3.2.2)
During the decade of the 1970s, Bransford and Johnson (1973) challenged the idea that
schemas work at retrieval stage. They constructed texts that described a situation in such a
way that the reader was unable to understand its meaning unless some clues were provided.
The researchers suggested that when new information cannot be related to an appropriate
schema, very little is remembered. Other researchers found similar results in comparative
experiments of prose retention (see Dooling and Lachman, 1971).

Today two kinds of schemas are distinguished: event schemas and story schemas. Event
schemas consist of knowledge about the subject matter of the story (Cohen, 1989; Christy et
al, 2017). For example, the event schemas activated in remembering *The Man Who Mistook*
*His Wife for a Hat* by Oliver Sacks (a collection of different narratives about Oliver Sacks'
patients) might include knowledge of psychiatric hospitals, admire characters, self-identity,
physiology of the brain, sensory ghosts, disembodiment, or autism. Story schemas consist of
abstract, content-free knowledge about the structure of a typical story.

For science communication, one of the advantages of story schemas is that the majority of
people have been exposed to them since childhood in such forms as religious instruction,
drama, or reading fictional literature. Therefore it represents a widespread and well-
established knowledge held by the general public that can be used, without previous
instruction, to the benefit of popularisation of science.

***Final note***

It is still necessary to invest considerable amount of effort to investigate about the use of
narratives in science communication as it is a rather recent field as well as a promising one.
For instance, it is necessary to explore in more depth the adequate characteristics of narrative
text for effective science communication (i.e. the use of powerful mnemonic devices). From
my perspective, science communication via narratives should follow a series of rules, as it
happens with other narrative sub-genera such as the thrillers, horror stories, historic novel,
etc.  I have named this kind of narratives "SciComm narratives" (Negrete, 2014) and they
could be considered as a new narrative sub-genera with their own characteristics and rules.
Therefore, it is important to generate more knowledge that enables us to provide a solid
theoretical body around narratives for science communication (SciComm narratives).

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
