# Peer review of "Remember rhythm and rhyme: Memorability of narratives for science communication"

_Geoscience Communication, 2020_

## Referee Comment (RC1) · Sara J ElShafie (Referee) · 18 Jul 2020

**GENERAL COMMENTS**

I think that an article explaining how narratives improve memory and thus aid science communication will be incredibly valuable. I'm happy to see this work in progress and I look forward to citing it in my own work.

However, this draft is mostly about memory and contains very little on narrative. For a paper entitled "Memory and narratives in science communication," the discussion should be more balanced. I would like to see narrative defined and cited as rigorously as memory is treated in the current draft. I also think this work would benefit from introducing narrative earlier in the piece and incorporating discussion of how memory

and narrative interplay throughout the manuscript.

This paper would be further strengthened by using actual narratives throughout to illustrate the author's points. For example, toward the beginning of the manuscript, offer the reader a list of facts (e.g., definitions of jargon related to the study of memory) and then tell the reader a story (e.g., about someone who needed to remember something). Later in the manuscript, ask the reader to recall the definition of one of the jargon terms presented early on. Then ask the reader to recall a specific incident from the story told earlier. Which was easier to recall? This will demonstrate the author's thesis!

Adding some figures, especially to illustrate some of the jargon or experiments described in the literature review, would also help to make this paper more engaging and accessible.

SPECIFIC COMMENTS

Line 1: Did the author intend to use the word "rhyme" rather than "rime" in the title? I'm guessing the former, since the author uses the word "rhyme" several times throughout the manuscript. While the word "rime" is a term used in literature, the word "rhyme" seems more relevant to this paper.

Lines 12-15: This is a very long sentence. I recommend breaking it up.

Lines 16-24: I suggest rephrasing the third sentence in the abstract, and each sentence following, as a statement rather than a mention of something that will be discussed. For example, "Evidence from the field of memory studies suggests that narratives represent a good recall device" rather than "I will present evidence. . ."

Lines 132-133: Please explain this experiment. A figure showing the potential curve would help.

Line 137: Citation needed.

Lines 138-140: Please explain this work.
Lines 140-143: This is a really important and interesting example. I recommend elaborating on the role of story in geoscience knowledge across different cultures and offering some specific examples.

Lines 165-167: Citation(s) needed.

Line 243: Citation needed.

Lines 245-249: Put this example after the following paragraph.

Lines 260-166: The specific examples offered here help me understand the author's point and remember the information later (because those examples are themselves mnemonics!). I recommend using specific examples throughout the manuscript in this way, and especially to use examples of how any given memory concept applies to science communication specifically.

Line 277-279: I recommend omitting the sentence in line 277 and instead introducing the next paragraph with a sentence like, "One way to achieve organization is to introduce objects together."

Lines 297-301: This sentence is long and difficult to digest. It is also redundant with the next paragraph. I recommend omitting it. Furthermore, it seems to me that the discussion of Models in section 3.9.2 should precede the section on Story schemas. I suggest adding the first sentence under 3.9 to the first paragraph under 3.9.2 and making that section "3.9 Models." Then "Story schemas" can be its own section (3.10).

Lines 326-327: A brief description of this story would be helpful here for those unfamiliar with it. The story also needs a citation.

Lines 328-329: Please give an example.

Line 343: It is more inclusive to use the pronouns "they/them/themselves" when referring to a hypothetical person.

Lines 367-421: Following my general comments, the entire section entitled "Narratives

as mnemonic devices for science communication" should be broken up and incorporated throughout the manuscript. The author has already indicated exactly how they can do this – just take each paragraph and incorporate it into the section(s) referenced within that paragraph. Also, the entire commentary on "Narratives as mnemonic devices..." needs citations. I do not see a single citation in this section until the last paragraph when a few examples of stories are cited.

Lines 415-416: Please rephrase to be clearer.

Line 416: Carbon by Primo Levi is listed without justification. Please either clarify what this story exemplifies, or omit it.

Lines 420-426: This sentence is awkward and redundant. I recommend omitting it and instead ending that paragraph with the last sentence of the paper (which does not need a separate subheading).

TECHNICAL CORRECTIONS

"Short-term" and "long-term memory" should always have a hyphen before "term." The hyphen is missing in several instances (e.g., line 197). Please search for "short term memory" and replace with "short-term memory," etc.

The author alternates between American and British spellings of words. For example, "organisation" is spelled with an "s" in line 276, but is then spelled with a "z" in the next sentence in line 277. Please review the manuscript and ensure that all spellings are consistent with whichever style the publishing journal uses.

Please also see the attached manuscript file with typos highlighted.

Thank you for the opportunity to review this work! I enjoyed reading it and learned quite a bit. I think it will be a valuable resource for anyone interested in communication, cognition, storytelling, narratology, education, and anything related!

Please also note the supplement to this comment:

https://gc.copernicus.org/preprints/gc-2020-20/gc-2020-20-RC1-supplement.pdf

---

## Author Comment (AC1) · 26 Jul 2020

Thank you very much for your comments, they are very interesting and pertinent. I will proceed to make the necessary amendments in my paper. I am positive that with this refinements my work will improve substantially. I will comment more as I progress in the edition. Thanks again!

---

## Referee Comment (RC2) · Sara J ElShafie (Referee) · 10 Aug 2020

Hello Aquiles, Thank you for updating me on your changes. I am so glad that you found my suggestions helpful! Thank you again for the opportunity to review your work. I look forward to seeing the final version of your paper when it is published! Best regards, Sara

---

## Referee Comment (RC3) · Nancy Longnecker (Referee) · 10 Aug 2020

Review: gc-2020-20

Remember rhythm and rime*: Memory and narratives in science communication

*Should this be 'rhyme'?

GENERAL COMMENTS I enjoyed reading this manuscript which provides a useful historical overview of early work that developed understanding of how humans remember. This is of fundamental importance in science communication. The value of stories in science communication is an area that warrants much more research and this manuscript should encourage that endeavour.

[Figure]

I have marked up the manuscript pdf with some minor copy edit suggestions.

My main recommendation is to update the manuscript with more current findings. Has anything been done that builds on this historical work? As I am not a cognitive psychologist, I cannot point out specific studies or new concepts, but I would be surprised if equipment that is now available and new research methods and ideas have not been used to elucidate some aspects of memory. For example, is there any work with tracking eye movements to measure engagement with story and then looking at recall? (I have no idea, but it would be interesting, and someone may have done something along these lines.) With all of the work on memory decline in ageing populations, it seems likely there would also be relevant information in neuroscience literature.

As it is, except for a handful of citations from 2000, the only cited references to studies done within the last 20 years are those of the author. Surely there is relevant work in this space. If not, that in itself is noteworthy and should be mentioned.

The list below includes a few references which the author may already have. While some of them may be tangential to the points the author wants to make, they could be used to elaborate the last section about narratives and science communication and would help by including more recent work.

Braund, M., Ahmed, Z., 2018. Drama as physical role-play: actions and outcomes for life science lessons in South Africa. Journal of Biological Education 53, 412-421. Cormick, C. (2019). 'Who doesn't love a good story? — What neuroscience tells about how we respond to narratives'. JCOM 18 (05), Y01. https://doi.org/10.22323/2.18050401. Dahlstrom, M.F., 2012. The Persuasive Influence of Narrative Causality: Psychological Mechanism, Strength in Overcoming Resistance, and Persistence Over Time. Media Psychology 15, 303-326.

Dahlstrom, M.F., 2014. Using narratives and storytelling to communicate science with nonexpert audiences. Proc. Natl. Acad. Sci. U. S. A. 111 Suppl 4, 13614-13620. Gottschall, J. 2012. The storytelling animal: How stories make us human. Houghton

Mifflin Harcourt. Katz, Y. 2013. Against Storytelling of scientific results. Nature Methods. 10: 1045. Martin, K. M., Davis, L. S. and Sandretto, S. (2019). 'Students as storytellers: mobile-filmmaking to improve student engagement in school science'. JCOM 18 (05), A04. https://doi.org/10.22323/2.18050204. Martin, K. & Miller, E. 1988. Storytelling and science. Language Arts Vol. 65, No. 3, Literary Discourse as a Way of Knowing pp. 255-259 https://www.jstor.org/stable/41411379?seq=1 Olson, R. (2015). Houston, we have a narrative. Chicago, IL, U.S.A.: University of Chicago Press.

---

## Referee Comment (RC4) · Nancy Longnecker (Referee) · 10 Aug 2020

[referee-annotated manuscript omitted]

---

## Referee Comment (RC5) · Sara J ElShafie (Referee) · 10 Aug 2020

I agree with this suggestion about references. My own paper on this subject includes some citations on neuroscience and physiology research on how storytelling affects attention and recall. These might be of help. The paper itself might also be a useful reference to cite in the discussion of why we should use stories to engage the public with science. The paper is available open access with the following citation:

Sara J ElShafie, Making Science Meaningful for Broad Audiences through Stories, Integrative and Comparative Biology, Volume 58, Issue 6, December 2018, Pages 1213–1223, https://doi.org/10.1093/icb/icy103

---

## Author Comment (AC2) · 10 Aug 2020

Dear Sarah,

I found very interesting and useful your comments. Thank you very much!

Here I will comment on each of them:

1) Narrative is now introduced and defined earlier in the document. 2) Rhyme is the right word, thanks! 3) Abstract was modified according to your suggestions 4) Bartlett's experiment is now explained. 5) The citations that were missing are now included 6) Paragraphs were organized according to your comments 7) Models is a section now on its own 8) The section on Narratives as mnemonic devices was broken up and incorporated throughout the manuscript (this was the original arrangement but

the previous referee suggested to change it and create the section on Narratives as mnemonic devices). 9) Short-term and Long-term were corrected. 10) For me the problem regarding British and American English is confusing, as I am not a native speaker. I use normally British corrector (because I did my PhD in Britain). For British English the corrector accept both: organization and organisation. I checked the rest of the document with the British option and I changed the few spelling mistakes that the corrector marked as incorrect, but I am not sure if there are more mistakes. 11) The idea of introducing an interactive "exercise" with the reader is really interesting! However I did not write this paper with this idea in mind, I believe I will, no doubt, do this in the future but I need to create the paper right from the beginning with this intention in mind. Thanks again, is a wonderful idea making the paper a ludic one! 12) My experience with figures has been rather difficult because the ones available in the internet, most of them have copyright and I currently do not have access to a designer to create my own.

Thanks again for your comments. In my opinion the paper improved significantly thanks to your help!

Please also note the supplement to this comment:
https://gc.copernicus.org/preprints/gc-2020-20/gc-2020-20-AC2-supplement.pdf

---

## Referee Comment (RC6) · Nancy Longnecker (Referee) · 11 Aug 2020

Kia ora Sara,

What a great paper! Thank you for sharing that. Yes, Aquiles, the references cited in Sara's paper will be very useful for updating your manuscript. It is so nice to see this important work progressing.

Kind regards, Nancy
* * *

---

## Author Comment (AC3) · 29 Aug 2020

Thank you for your generous feedback!

I have corrected all the grammar and spelling mistakes that you marked as well as the missing bibliography.

I will proceed to look for the bibliography that you suggested.

I really appreciate all the comments that you sent me to improve my work, its looking better and better!

---

## Author Comment (AC5) · 29 Aug 2020

I will proceed to look for the bibliography that you suggested.

---

## Author Comment (AC8) · 15 Sep 2020

Final response

Changes regarding Dr. Sara ElShafie comments:

1) Narrative is now introduced and defined earlier in the document. 2) Rhyme is the right word, thanks!  3) The abstract was modified according to your suggestions 4) Bartlett's experiment is now explained.  5) The citations that were missing are now included.  6) Paragraphs were organized according to your comments 7) "Models" is a section now on its own 8) The section "Narratives as mnemonic devices" was broken up and incorporated throughout the manuscript (this was the original arrangement but the previous referee suggested to change it and create the section "Narratives as

mnemonic devices"). 9) "Short-term" and "Long-term" were corrected. 10) For me the problem regarding British and American English is confusing, as I am not a native speaker. I normally use British corrector (because I did my PhD in Britain). For British English the corrector accepts both: "organization" and "organisation". I checked the rest of the document with the British option and I changed the few spelling mistakes that the corrector marked as incorrect, but I am not sure if there are more mistakes. 11) The idea of introducing an interactive "exercise" to the reader is really interesting! However I did not write this paper with this idea in mind, I believe I will, no doubt, do this in the future but I need to create the paper right from the beginning with this intention in mind. Thanks again, it is a wonderful idea making the paper a ludic one! 12) My experience with figures has been rather difficult because most of those available in the internet have copyright and I currently do not have access (due to the quarantine) to a designer to create my own. General comments:

• The document is now reorganised in the following sections:

1. Abstract (there was no abstract in the previous version) 2. Introduction 3. Objective and methodology (this section did not exist in the previous version) 4. Literary review on memory studies (this section was shortened and refined) 5. Narratives as mnemonic devices for Science Communication (this is a new section required by the referees. It concentrates the view of the author about the importance of narrative mnemonic capabilities for science communication). 6. Final note 7. Bibliography

• Several technical words were defined, while others were omitted from the document in order to offer a better reading for non-specialists. • All the references related to brain structures were eliminated, they resulted too technical and unnecessary (therefore no brain diagram was needed). • I have corrected all the grammar and spelling mistakes marked by the referees • Missing bibliography was completed. • I revised the suggested literature and also included more recent quotes that I found on "Memory and Narratives" (Dr. Nancy Longnecker). I believe my paper looks much better now after reworking it with the referees' changes. Finally I would like to thank for all

the comments and interest showed in the interactive discussion phase.

Dr. Aquiles Negrete Yankelevich

---

## Author Response (AR1)

Here is a list of changes in the document:

1)  Now the paper contains a shortened version of Primo Levi's short story (*Nitrogen)* to illustrate the points made in the text and as an example of episodic memory. There are references to this narrative from different parts of the text (rhyme, rhythm, humour and images) to provide an example of memorable devices of narratives.

2)  The title was slightly modified :

    *Remember rhythm and rhyme:*
    *Memorability of narratives for science communication*

    I believe this title is more appropriate for the content of the paper, which concentrates on why narratives are memorable and not on how narratives and memory interact.

3)  A section was included that briefly explains empirical evidence (presented in some of my previous articles) 
[revised manuscript text omitted]

---

## Editor Decision (ED1)

**Gc-2020-20 DECISION**

The article is too theoretical and not easy to follow for those not in the field, and we are in a journal of Geoscience communication, where the potential readers are mainly unfamiliar with cognitive psychology. Following Referee 1 (Sara J ElShafie ), I believe the article would benefit by using actual narratives throughout to illustrate the author's points, even not in a ludic way, but simply to give examples. This has not been accomplished yet. While I believe it has been a good choice to introduce narrative earlier in the paper, at the same time incorporating discussion of how memory and narrative interplay through the manuscript does not help the inexperienced reader. The reader not in the field risk losing the overview. So I suggest, to give a summary on how narrative and memory interplay and how this can help science communication, at the end of the article before the final discussion or in the final discussion itself.

I also believe it would be advisable as suggested by Referee 1 elaborating on the role of story in geoscience knowledge across different cultures and offering some specific examples. Or at least make more references to "geo-stories".

All in all, even if in par.2 the author indicates that the objective of his work is to provide a literary review of memory regarding memorability of narratives, the review results too long. In accordance with Referee 1 I must remark that the paper still contains very little on narrative. I believe that using examples of narrative will help the author and the reader to share the content of the article. In other words, the author should use narrative to accomplish what it is discussed in the paper itself: communicating in an efficacious way through narrative.

Always in par. 2 the author refers to previous articles where he provided empirical evidence suggesting that narratives represent a memorable text format. Even if the interested reader can read these articles, I believe that making explicit reference to some empirical evidence would make this article more accessible to a wider audience.

Specific suggestions related to the present decision have been annotated in the last draft of the paper.